# Towards Precision Oncology: The Role of Smoothened and Its Variants in Cancer

**DOI:** 10.3390/jpm12101648

**Published:** 2022-10-05

**Authors:** Alina Nicheperovich, Andrea Townsend-Nicholson

**Affiliations:** Institute of Structural and Molecular Biology, Research Department of Structural and Molecular Biology, Division of Biosciences, University College London, London WC1E 6BT, UK

**Keywords:** Smoothened, cancer, precision medicine, therapy

## Abstract

The G protein-coupled receptor Smoothened (Smo) is a central signal transducer of the Hedgehog (Hh) pathway which has been linked to diverse forms of tumours. Stimulated by advancements in structural and functional characterisation, the Smo receptor has been recognised as an important therapeutic target in Hh-driven cancers, and several Smo inhibitors have now been approved for cancer therapy. This receptor is also known to be an oncoprotein itself and its gain-of-function variants have been associated with skin, brain, and liver cancers. According to the COSMIC database, oncogenic mutations of Smo have been identified in various other tumours, although their oncogenic effect remains unknown in these tissues. Drug resistance is a common challenge in cancer therapies targeting Smo, and data analysis shows that healthy individuals also harbour resistance mutations. Based on the importance of Smo in cancer progression and the high incidence of resistance towards Smo inhibitors, this review suggests that detection of Smo variants through tumour profiling could lead to increased precision and improved outcomes of anti-cancer treatments.

## 1. Introduction

The G protein-coupled receptors (GPCRs) are the largest family of transmembrane proteins encoded by 826 genes in the human genome [1]. These receptors play important roles in mediating a plethora of physiological responses. Structurally, all GPCRs are characterised by seven transmembrane α-helical regions separated by intra- and extracellular loops. Human GPCRs are split into four classes based on structure and sequence similarity: A (rhodopsin), B (secretin), C (glutamate), and F (Frizzled). Class F consists of Smoothened (Smo) and 10 paralogues of Frizzled, which regulate the Hedgehog (Hh) and Wnt signalling pathways, respectively [2]. Whilst class F remains one of the least structurally characterised subfamilies of GPCR receptors, recent structural insights into Smo have enabled scientists to elucidate certain aspects of its activation and regulation. These findings make Smo the only receptor of its class that has been established as a therapeutic target to date [3].

Deregulation of the Hh pathway has been implicated in a broad spectrum of tumours including cancers that form in the liver, breast, pancreas, and skin [4,5,6,7]. In many of these, Smo was recognised as a promising drug target [6,8,9,10]. In some types of cancer, in fact, Smo drives tumour progression [8,10,11,12,13]. For this reason, Hh-focused drug discovery efforts have predominantly been dedicated to targeting this receptor. Indeed, three small-molecule antagonists of Smo have now been approved by the US Food and Drug Administration (FDA) [14]. However, Smo is the second most variable member of class F GPCRs, with over 200 variants identified in the healthy population [15]. Some of the mutations render Smo resistant to inhibition by therapeutic antagonists thus impairing quality of life through undesirable side effects and making cancer therapy ineffective. Although it has been recognised that polymorphisms of Class A GPCRs may have profound implications for precision medicine [16], researchers are only beginning to evaluate the role of class F receptors in patient-specific treatment approaches. This review attempts to show that, considering the mutational landscape of Smo and its role in cancer, identification of Smo mutations through tumour genetic screening can enhance diagnosis and guide the choice of medication to ensure the best therapeutic outcomes for cancer patients.

## 2. The Structure of Smoothened

### 2.1. Domain Architecture and Binding Pockets

Human Smo is a 787-residue transmembrane protein encoded by the *SMO* gene with a structure characteristic of class F GPCRs [17]. It has an extracellular N-terminus segment which contains a cysteine-rich domain (CRD), followed by the linker (LD) and hinge (HD) domains. The heptahelical transmembrane domain (7TM) consists of 7 transmembrane helices (TM1-7) that are separated by alternating extracellular and intracellular loops (ECLs and ICLs). The flexible C-terminal domain is found on the intracellular side of the membrane (Figure 1a,b) [18].

Structural studies of Smo have provided clues to the mechanisms underlying Smo regulation. Smo activation is negatively regulated by a 12-pass transmembrane receptor Patched (Ptc). It was originally suggested that the inhibitory effect of Ptc was mediated by direct binding to Smo [19], although a significant amount of contradicting evidence invalidated this view [20]. It is now established that Ptc exerts its inhibitory effect by limiting the accessibility of lipid-based agonists such as cholesterol and oxysterol in the local membrane environment of Smo [21]. Structural studies have provided empirical evidence for this mode of regulation. For example, a recent crystal structure of Smo clearly shows three ligand-binding pockets of the receptor, two of which are occupied by cholesterol (Figure 1c) [22]. The latest cryo-electron microscopy (cryo-EM) structure of lipid-bound Smo also confirms that lipid binding is a major regulator of the receptor’s activity [23]. Although lipid binding alone is enough for Smo to acquire its active conformation, another cryo-EM structure has shown that ligand binding can also induce receptor coupling to a heterotrimeric G-protein, G_i_ [24]. This suggests that Smo can mediate downstream signal transduction through G_i_-dependent and G_i_-independent mechanisms. Other crystal structures present a variety of binding states of Smo, including conformations where a ligand is only bound to the CRD [25], both CRD and upper 7TM binding sites [26], or the upper 7TM only [27,28,29]. Thus, the emerging structural evidence highlights the complexity of Smo activation, and the presence of multiple ligand-binding pockets highlights this receptor’s inherent druggability.

### 2.2. The Molecular Switch Mechanism of Activation

Strict regulation of Smo activity is essential to avoid aberrant signalling through the Hh pathway. Although members of Class A GPCRs have well-characterised motifs that play a key role in regulation of receptor activity, the lack of these segments in Smo has delayed elucidation of its molecular switch mechanism until recently.

A large-scale sequence alignment of mammalian and non-mammalian class F receptors has identified a conserved basic residue R451 in the TM6 of Smo. Analyses of available structures combined with computational modelling have shown that this residue acts as a molecular switch through interactions with the TM7 [30]. In its inactive state, R451 forms a network of interactions with residues T534, W535, and W537 found in the lower part of the TM7. In the lipid-bound active conformation, however, these interactions are weakened due to the substantial rearrangement of cytoplasmic regions of the transmembrane helices, particularly in TM5 and TM6 (Figure 2a) [30]. Indeed, the previously mentioned cryo-EM structure of G protein-bound Smo shows that the switch from the inactive to active states is accompanied by an increase in the distance between R451 and W535 by 0.5 Å [24]. This movement is required to accommodate the α5 helix of the G_i_α subunit in order to allow the guanine nucleotide exchange activity of the receptor [31] (Figure 2b).

Examination of structural rearrangements reveals that the Smo receptor exists in distinct active and inactive conformational states. Interestingly, certain oncogenic mutants such as Smo-W535L mimic the active state to trigger ectopic Hh signalling [10]. The wide variety of structures adopted by Smo could thus facilitate discovery of biased drugs that would target this receptor in a conformation-selective manner.

## 3. Hedgehog Signalling and Regulation of Smoothened

### 3.1. Smoothened and the Hedgehog Pathway

Smo is a key player of the Hh pathway which is essential in many physiological processes including tissue patterning, regeneration, and homeostasis [32]. In mammals, Hh signalling is activated by small Hh proteins Sonic, Desert, and Indian [33]. Binding of these ligands to Ptc can induce target gene expression, which is mediated by members of the glioma-associated oncogene (Gli) transcription factor family, via the canonical pathway. Alternatively, Hh ligands can generate a response in a Gli-independent fashion through the non-canonical pathway. Non-canonical signalling can be split into two signalling modes: Smo-independent (type I) and that downstream of Smo (type II) [34]. Since the focus of this review is the Smo receptor, only the canonical and Smo-dependent non-canonical pathways will be considered further.

In the absence of Hh ligands, the levels of Smo in the plasma membrane are low and Ptc indirectly inhibits Smo activity by altering the lipid composition of the cell membrane [21]. Binding of a Hh ligand to Ptc leads to the release of this inhibition. Further activation of the pathway requires translocation of Smo receptors sequestered in the intracellular vesicles to the cell membrane [34]. Thus, prior to the start of Hh signal transduction, inhibition of Smo by Ptc is abolished and the GPCR is enriched in the cell surface membrane.

In the canonical pathway (Figure 3a), Smo indirectly mediates transcription of Hh target genes through the Gli family of transcription factors. Firstly, increased membrane localisation of Smo receptors is followed by phosphorylation of their C-terminal tails by casein kinase 1α (CK1ɑ) and G protein-coupled receptor kinase 2 (GRK2) [35]. This causes conformational rearrangements within the receptor which enable it to laterally move to the membrane of primary cilia (PC), long microtubule-based organelles that protrude from the cell [36]. This translocation event is essential for canonical Hh signalling in vertebrates [37]. In the PC, Smo interacts with the Ellis-van Creveld syndrome protein complex (Evc/Evc2) via its phosphorylated C-tail. This triggers the dissociation of the inhibitory suppressor of fused homolog (SUFU) from Gli, resulting in the release of the transcription factor [38]. Gli then enters the nucleus, where it gets converted into its transcriptional activator form GliA which induces transcription of the Hh target genes [39]. In addition to beneficial roles such as adult organ homeostasis and repair [32], these genes play a role in angiogenesis, metastasis, and chemotherapy resistance [14], as discussed in the following chapter of this review.

In contrast to the canonical pathway, type II non-canonical Hh signalling (Figure 3b) elicits a more rapid cellular response without any detectable Gli activity [40]. In this pathway, Smo acts as a GPCR, i.e., it mediates the effects of Hh ligands through interactions with the heterotrimeric G protein G_i_. When Ptc is bound by a Hh ligand, activated Smo exchanges GDP for GTP in G_i_, resulting in the release of the Gβ𝛾 subunit. The dissociated Gβ𝛾 subunit induces phospholipase C-𝛾 (PLC𝛾)-catalysed synthesis of inositol triphosphate (IP_3_), which triggers Ca^2+^ release from the endoplasmic reticulum (ER) resulting in calcium spikes [41]. Interestingly, this signalling cascade is the only axis of the non-canonical pathway that requires accumulation of Smo in PC [42]. Additionally, Gβ𝛾 activates phosphoinositide 3-kinase (PI3K), which in turn activates the Rho family small GTPases RhoA and Rac1 that modulate the actin cytoskeleton, thereby promoting migration and fibre formation [43,44]. Considering the fundamental kinases that are involved in this transduction network, it is not surprising that G protein-mediated signalling through Smo tightly interlinks with other pathways. These include Ras-Raf-MEK-ERK and PI3K-AKT-mTOR, increased activity of which has been associated with the development of Smo inhibitor resistance and tumour evolution [45,46].

A recently identified Gli-independent signalling cascade presents another piece of evidence that highlights the importance of non-canonical signalling in cancer. It has been shown that Smo-mediated dissociation of G_i_ contributes to activation of nuclear factor κB (NF-κB) in diffuse large B-cell lymphoma cells. This is achieved through the recruitment of CARD recruited membrane-associated (CARMA), B-cell lymphoma/leukaemia 10 (BCL-10), and mucosa-associated lymphoid tissue 1 (MALT1) proteins, which triggers a release of NF-κB inhibition [47]. This is followed by the translocation of cytosolic NF-κB into the nucleus, where it drives the expression of anti-apoptotic and pro-proliferative target genes that can enhance cancer cell survival [48]. Thus, crosstalk between the Hh and other signalling pathways can amplify the effect of dysregulated Smo-dependent signalling, thereby driving cancer development and progression.

### 3.2. Regulation of Smoothened Activity

The observation that Smo has important functions outside of its GPCR-like interactions with G_i_ has encouraged thorough research into the mechanisms underlying regulation of this receptor. Post-translational modifications are now known to regulate Smo activity, with phosphorylation, N-glycosylation, ubiquitination, and recently discovered cholesterylation having particular regulatory importance, although sumoylation has also been described [49]. Moreover, the binding of small lipid-based molecules has an important regulatory effect on Smo. Alongside protein–protein interactions with G_i_, Smo has been found to oligomerise, interact with β-arrestin and recently with G_12_ [50,51,52], but these aspects of Smo regulation are outside the scope of this review.

#### 3.2.1. Post-Translational Modifications

Phosphorylation of the C-tail of Smo by CK1α and GRK2 is an essential activation step that induces a conformational change of the receptor and its subsequent enrichment in the PC membrane [35]. In vitro kinase assays with murine Smo led to the identification of specific serine/threonine motifs where phosphorylation takes place, and sequence alignment has shown that these residues are conserved in the human counterpart (Figure 4). Additionally, the same study suggests that an oncogenic form of Smo (R562Q) exhibits increased phosphorylation and basal activity by making the C-terminal tail more accessible to phosphorylation by kinases [35].

Another post-translational modification that occurs in Smo is *N*-glycosylation. Both the N-terminal segment and the extracellular loops harbour *N*-glycosylation sites (Figure 4) [53]. Experiments involving murine Smo have shown that although mutations in these sites have no effect on PC trafficking and canonical signalling, they lead to a diminished ability of Smo to induce non-canonical signalling via G_i_. Based on these results, it was suggested that the removal of *N*-linked glycosylation may lead to certain perturbations in the structure that make the receptor adopt a conformation biased towards canonical signalling [53]. Indeed, there is an increasing amount of evidence indicating that *N*-glycosylation affects ligand binding in several other members of the GPCR superfamily [54,55]. Given that some of the glycan acceptor sites are proximal to the CRD and upper 7TM pockets of Smo, it is likely to also be true for this class F receptor. Further elucidation of the role of this modification in Smo regulation will aid the development of biased therapeutic ligands.

Latest research provides significant evidence for the role of the ubiquitination state of Smo in signalling. Ubiquitination of lysine residues on the cytoplasmic surface of Smo has been implicated in the removal of inactive Smo from PC when Hh signalling is suppressed. In fact, a mouse mutant Smo lacking all lysine residues on its intracellular surface was associated with increased accumulation in cilia and amplified levels of Hh target gene expression. More specifically, conserved lysine residues in the IL3 (corresponding to K430, K440, and K444 in human) were found to be critical for this Hh-independent ciliary accumulation (Figure 4) [56]. Therefore, a mutation in one of these residues or nearby regions could dampen ubiquitination and thus removal of Smo from PC, thereby causing pathway activation. Interestingly, one of the previously mentioned oncogenic forms of Smo (W535L) has a mutation near the cytoplasmic surface. This mutant leads to increased receptor accumulation in PC and constitutive activation of the Hh pathway [10]. It appears this mutant mimics the active conformation of the receptor and is therefore not a suitable substrate for ubiquitination by a ubiquitin ligase [56].

Although it is well established that Smo is positively regulated by transient interactions with lipid-based agonists in its binding pockets, a recent study revealed an unexpected covalent attachment of a cholesterol molecule in the CRD of Smo in the presence of Hh ligands. Mass spectrometry analysis indicated that cholesterylation occurs on D95 (Figure 4). The authors have also demonstrated that loss of this modification failed to activate downstream signalling and reduced ciliary localisation [57]. In a more recent study, the same group confirmed the importance of this modification by observing that mice lacking the cholesterylation site were not able to survive past the embryonic stage [58]. Given the role of this modification in Smo regulation, interfering with cholesterylation of Smo may provide a novel therapeutic avenue to treat Hh-driven cancers.

#### 3.2.2. Small Molecules

Since it was discovered that Ptc inhibits Smo in a catalytic manner rather than by direct protein–protein interaction, multiple lines of evidence suggested that lipid-based small molecules are in fact endogenous ligands of Smo. These activating small molecules include cholesterol and oxysterol, and both have been shown to be efficient for cellular activation of Smo in the absence of Hh ligands [26,59]. At first, crystal structures showed that cholesterol and oxysterol individually bind the ligand-binding pocket in CRD [25,60]. More recent structures revealed that these lipid modulators can also bind the 7TM core of Smo [22,23], which explains why CRD-devoid Smo still gets activated in response to Hh [61]. Elucidation of the structural basis of Smo modulation by small molecules has enabled the identification of druggable pockets and key residues, which, if mutated, can alter the ligand binding affinity of the receptor.

In addition to positive regulation by lipid-based ligands, Smo activity was found to be negatively modulated by binding of the natural sterol alkaloid cyclopamine [62]. Like oxysterol and cholesterol, this molecule can bind both the CRD and the TM7 core [63]. Originally, this molecule was considered to have exclusively inhibitory effects on Smo, which was supported by decreased Hh gene expression in breast cancer and hepatocellular carcinoma (HCC) cells in response to cyclopamine [64,65]. However, it was also observed that cyclopamine can induce Smo accumulation in PC [36], indicating agonist-like properties of this compound. Nevertheless, the characterisation of Smo inhibition by cyclopamine led to the development of antagonistic cyclopamine derivatives with increased specificity and pharmacological potency. These include the FDA-approved Smo inhibitors used to treat Hh-driven cancers as well as other synthetic agonists that are currently undergoing clinical trials. These will be considered in the following chapter of this review.

## 4. Targeting Smoothened in Cancer

### 4.1. The Role of Hedgehog Signalling in Cancer

Although Hh signalling is active predominantly during development, this pathway plays a crucial role in adult tissue maintenance and repair [32]. However, it is becoming increasingly clear that aberrant activation of Smo-mediated signalling can disrupt these processes and trigger tumorigenesis. This is unsurprising since Gli upregulates expression of genes encoding proteins important in cell proliferation (cyclin D, cyclin E) [66], cell survival (Bcl-2) [67], angiogenesis (VEGF, Ang-1, Ang-2) [68], metastasis (MMP2, Snail) [69,70], and regulation of the replicative potential of the cell (TERT) [71]. Moreover, Smo-dependent non-canonical signalling can rewire cellular metabolism by increasing cellular Ca^2+^ levels, which in turn activates AMP-activated protein kinase (AMPK), a key sensor of cellular energetics. This leads to a rapid shift from oxidative phosphorylation to aerobic glycolysis [42], a well-established feature of cancer.

Besides the elevated expression of Gli targets and Ca^2+^ oscillations, signalling downstream of Smo can manifest cancer characteristics in an indirect manner. For example, in vivo experiments have demonstrated that Hh-driven basal cell carcinoma (BCC) cells have decreased accumulation of p53, a key growth suppressor that triggers cell cycle arrest [72]. Aberrant Hh activity has also been suggested to elevate DNA damage responses via crosstalk with the ATR/Chk1 pathway [73]. This was confirmed by the observation that a Smo inhibitor re-sensitises hematopoietic cells to UV-induced bulky DNA lesions [74]. Notably, this study highlights the direct involvement of Smo in DNA damage resistance, one of the main mechanisms underlying genome instability in cancer. Moreover, inhibition of Smo in BCC cells has been linked to increased levels of the major histocompatibility complex (MHC) Class I molecules [75], surface recognition elements that attract cytotoxic T-cells to tumours. This indicates that constitutive Hh signalling can enable cancers to evade the immune system by dampening cell surface presentation of these important proteins. Finally, although the role of Hh in inflammation is unclear, recent evidence suggests that this pathway plays a role in the initiation of gastric inflammation through elevated *SLFN5* expression, which can eventually lead to tumorigenesis in the foregut [76]. Smo can also indirectly stimulate tumour-promoting inflammation by activating NF-κB through the previously described non-canonical signalling axis [48]. Thus, multiple lines of experimental evidence indicate that dysregulated signalling through Smo can lead to major cellular alterations underlying the hallmarks of cancer (Figure 5) [77].

It is thus becoming increasingly clear that Hh signalling has a crucial role in tumorigenesis. Recognition that aberrant Hh signalling can be induced in several ways has led to classification of tumours based on the mechanism of Hh pathway activation. Type I cancers arise independently of the ligand due to activating mutations in *SMO* or inactivating mutations in *PTC* or *SUFU*. Type I tumours harbouring Smo mutants include BCC, basal cell nevus syndrome (BCN), medulloblastoma, meningioma, and HCC, and these will be described in more detail in the next chapter. Type II mode of activation is ligand-dependent and autocrine, meaning that the tumour cell releases Hh ligands which act upon itself. Type III tumour development is also ligand-dependent but paracrine, i.e., Hh ligands are released by stromal cells surrounding the malignancy [78]. Thus, inhibition of the Hh pathway is a very attractive therapeutic approach for tumours that fall under this classification system.

**Figure 5 jpm-12-01648-f005:**
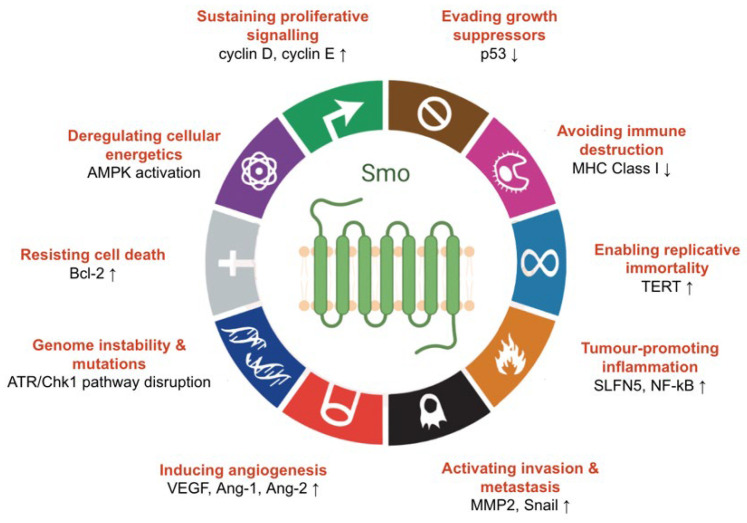
Smo-mediated Hh signalling facilitates the hallmarks of cancer. A compelling body of evidence indicates that Smo can contribute to cancer development through canonical signalling (sustained proliferative signalling, evasion of cell death, angiogenesis, invasion, metastasis, and replicative immortality), type II non-canonical signalling (deregulation of cellular energetics, tumour-promoting inflammation by NF-κB activation), or crosstalk with other pathways (genome instability, mutation accumulation, evasion of immune destruction and growth suppressors, aberrant inflammation due to increased SLFN5 levels) (modified from [77]).

### 4.2. Smoothened as a Therapeutic Target

As Type I cancers arise independently of the ligand, inhibition of ligand binding to Ptc will have no effect on this type of tumours. Therefore, inhibition must take place at the level of Smo or downstream of the receptor to have therapeutic value. Despite the currently active research into Gli inhibitors, the intracellular location of these transcription factors makes the drug discovery efforts more challenging. In contrast, Smo is found on the cell surface membrane, which facilitates interactions with drug molecules found in the extracellular milieu. Furthermore, Smo has several druggable pockets, as described in Section 2.1, which can facilitate the design of a diverse range of synthetic antagonists. Finally, Smo is an actively studied GPCR and complete elucidation of its activation mechanisms will eventually enable the discovery of highly specific and efficient therapeutics. Thus, Smo presents an important class of Hh inhibitors. Importantly, since Smo inhibition would hinder downstream activation of Gli and thus block expression of tumour-associated target genes, Smo antagonists can benefit patients with all three types (I, II, and III) of Hh-driven cancers.

As of February 2022, three Smo antagonists have been approved by the FDA: GDC-0449 (vismodegib) and LDE-225 (sonidegib) for the treatment of BCC, and PF-04449913 (glasdegib) for acute myeloid leukaemia [14]. Vismodegib, sonidegib, and glasdegib prevent Smo activation by blocking the binding sites of lipid-based agonists, which are found at increased levels in the plasma membrane in the presence of Hh. These small molecule inhibitors are also being investigated for the treatment of other cancers. For example, vismodegib has shown promising results in patients with BCN and medulloblastoma [79,80], and is currently undergoing trials for the treatment of pancreatic, colorectal, prostate, and breast cancers [81]. Sonidegib trials have also shown encouraging results in individuals with breast cancer [82], and patients are currently being recruited to investigate the effect of this drug on various types of advanced solid tumours [83]. Moreover, there is active recruitment of patients for clinical trials evaluating the efficacy of glasdegib against advanced soft-tissue sarcomas [84]. Other therapeutic Smo antagonists have not yet been approved and are currently going through clinical trials, with over half of them undergoing Phase II trials (Table 1).

## 5. Smoothened Variants and Their Impact on Cancer Therapy

### 5.1. Prevalence and Incidence of Genetic Variation

Rapid advancements in genomics over the last decade have enabled scientists to identify mutations in drug targets and examine effects of these genetic changes on therapeutic outcome. The recognition of the GPCR superfamily as a group of highly variable receptors was brought by a recent analysis of genetic data from over 60,000 unrelated healthy individuals deposited in the Exome Aggregation Consortium (ExAC) [85], results of which are available on the GPCR database (GPCRdb) website [15]. Given that members of this receptor family are targeted by over 30% of the FDA-approved therapeutics [86], this finding is likely to facilitate a more personalised, genetics-guided approach to pharmacological inhibition of GPCRs.

As for Smo, research concerning the effects of its mutations in cancer therapies is in its preliminary stages. Smo mutations can be divided in the following groups: (1) passenger mutations that do not contribute to cancer progression; (2) oncogenic mutations that stabilise the active form of Smo and release it from the inactive state conformational constraints; (3) resistance mutations that prevent effective drug binding. According to the GPCRdb, mutations in the *SMO* gene are rare across the healthy population, i.e., there is no mutant allele with a frequency of ≥0.01. Nevertheless, Smo, which is the only class F GPCR recognised as a drug target, is the second most variable member of its class, with records of 220 variants present in the GPCRdb. Five of these variants are loss-of-function, and 104 were predicted to have a deleterious effect based on their SIFT and PolyPhen scores. Examination of the mutational landscape of *SMO* shows that regions flanking the 7TM domain are the most mutated, with 8 out of 9 the most frequently occurring mutations located in these segments (Figure 6a, top).

To compare the mutational landscape of *SMO* in healthy individuals and cancer patients, relative frequencies of mutations in this gene were calculated using data from the COSMIC database, which stores information on somatically acquired mutations found in human cancer [87]. The histogram (Figure 6a, bottom) indicates that the 7TM domain is more frequently mutated across cancer patients compared to healthy individuals. To account for the bias associated with domain lengths, the average number of mutations per residue in each of the individual transmembrane regions and the flanking segments was calculated for both datasets. Figure 6b confirms that the 7TM domain experiences a higher mutational pressure in the context of cancer, with the exception of TM2. Since this transmembrane region harbours ligand-binding pockets and plays a central role in the conformational rearrangements underpinning receptor activation, increased mutational frequency in 7TM raises the probability of altered receptor function that would favour oncogenesis. On the other hand, a vast majority of the non-silent *SMO* mutations are missense in nature in both datasets, although nonsense mutations appear to be more prevalent in cancer (Figure 6c).

In addition to the general statistics, the COSMIC database provides important information on the distribution of somatic mutations in SMO across different tissues. According to the literature, a significant proportion (10%) of BCC tumours occur due to gain-of-function mutations in SMO, causing uncontrolled proliferation of skin cells [88]. On the other hand, alterations in the SMO gene have been described to be rare in chondrosarcoma [89] and gastric tumours [90]. Yet, according to the COSMIC database, mutations in the SMO gene have been identified in 35 out of 45 (78%) primary cancer tissue types, which highlights the importance of further investigation into the functional role of Smo mutants in various tumours. These genetic alterations are the most prevalent in meninges (12.8%), skin (7.91%) and uterine adnexa (7.69%) [87]. It is important to note, however, that most of these genetic alterations are passenger mutations, and the relatively high mutational frequency may be due to high genomic instability and proliferative potential of these tumour cells.

### 5.2. Activating Mutations

The role of Smo mutants in oncogenesis was first described in BCC, the commonest form of human cancer [10]. Direct sequencing of the *SMO* gene in BCC patients identified two oncogenic mutations. One of these, R562Q, is found in close proximity to the C-terminal Ser/Thr motif (S588, S590, T593, S595) that gets phosphorylated by CK1α. This genetic alteration causes conformational changes that make the C-tail more accessible for phosphorylation by the kinase, hence leading to increased activity of the receptor [35]. The other sporadic mutation changes codon 535 from tryptophan to leucine (W535L). As mentioned previously, this residue is a key player in the molecular switch mechanism of Smo activation. The W535L mutant localises to PC in the absence of Hh ligand [91] and has constitutive activity, as shown by increased mRNA levels of a Gli target transcript [10]. In addition, molecular dynamics simulations have shown that this mutation weakens the interactions between the cytoplasmic ends of TM6 and TM7, making these helices tilt outwards to allow coupling with G_i_α [30]. These lines of evidence suggest that W535L conveys the ability to drive tumour progression through both canonical and non-canonical Hh signalling.

While mutations in *SMO* have a well-established association with sporadic BCC, two more recent case reports identified another oncogenic Smo mutation in an inherited form of skin cancer, BCN [11,92]. This somatic mutation is caused by a substitution of leucine to bulkier phenylalanine at the codon position 412 (L412F). This mutation is located in the TM5 and the substitution for a bulkier amino acid enables the conformational change between TM5 and TM6 that confers the active state (Figure 2) [93]. Indeed, this mutant yields a much higher Gli promoter activity compared to the wild type, which indicates that this mutation leads to increased signalling through the canonical pathway. Although vismodegib treatment was successful in the first reported case of Smo-driven BCN, in the second case clinicians decided against this treatment because this mutation had been linked to vismodegib resistance in BCC [94].

Another form of epithelial tumour, HCC, was also linked to a mutation in Smo. The K575M substitution located in the C-terminus of the receptor was initially thought to interfere with the physical interaction between Smo and Ptc, thereby making Smo constitutively active [8]. However, now that it has been established that Ptc controls Smo activity in an indirect manner, the mechanism of activation of this Smo mutant needs to be explored further. Perhaps it has a similar mechanism of activation to R562Q since it resides proximately to the CK1α phosphorylation site.

In addition to cancers of epithelial origin, Smo mutants have been associated with brain tumours. In addition to W535L, the S533N mutation, also located in the TM7, was first identified in human cancer patients with medulloblastoma [13] and was confirmed to drive the formation of brain tumours in mice models [95]. Like other oncogenic mutations, S533N is believed to destabilise the architecture of the transmembrane receptor to promote the transition to the active state. Substitution of a small, uncharged serine for a positively charged arginine with a bulkier side chain probably disturbs the interactions between TM6 and TM7. In addition, whole-genome analysis of medulloblastoma tumours has identified another novel oncogenic mutation, S278I, located in the TM2 [96]. Furthermore, meningioma, the most common type of primary brain tumour, is in some cases driven by the previously mentioned L412F and W535L mutations [12]. Interestingly, these mutations were found to be more abundant in a particular type of this tumour, olfactory groove meningiomas, compared to other types. Additionally, it has been reported that patients with grade I tumours driven by mutated Smo had a significantly poorer prognosis compared to tumours with the wildtype receptor [97]. Thus, the mutational status of Smo can be used to determine which patients can benefit from pharmacological inhibition of this receptor. The Type I tumours that are driven by activating mutations of Smo are summarised in Table 2.

It is notable that oncogenic mutations of Smo identified to date are found in different structural regions: TM2 (S278I), TM5 (L412F), TM7(S533N and W535L), and the C-tail (R562Q and K575M). This indicates that the conformational state of Smo is stabilised by interactions spanning multiple domains. By disrupting the structural integrity of the inactive form of Smo, these mutations activate the Hh signaling and trigger tumorigenesis. For this reason, it would be expected that these mutations would primarily be found in cancer patients. Strikingly, according to the GPCRdb records, the R562Q mutation was present in 3 healthy individuals. This could be explained by lower genetic penetrance of this mutation. Alternatively, it is possible that cancer had not developed or was not identified in these individuals at the time of genetic data analysis. The absence of other oncogenic mutations in the healthy population, however, confirms the role of these mutations in carcinogenesis.

### 5.3. Resistance Mutations

Despite Smo being a promising therapeutic target for Hh-driven cancers, drug resistance is a common challenge. Tumour relapse has been reported in patients treated with vismodegib and sonidegib due to acquired resistance, however no resistance has been reported towards glasdegib, which has a different binding site to the vismodegib and sonidegib [98]. One of the major mechanisms of resistance to Smo inhibitors is disruption of drug-receptor interactions because of mutations in the 7TM domain.

Although the Smo inhibitor vismodegib is successful in treating BCN [99], only 48% of patients with advanced BCC respond to the drug [100], and further 20% of the BCC patients develop resistance within the first year of treatment [101]. Additionally, vismodegib treatment has shown considerable tumour shrinkage in a patient with medulloblastoma but was followed by relapse within several months because of acquired resistance [102]. According to the COSMIC database, mutations in 12 residues have been attributed to vismodegib resistance (Figure 7a), and over a quarter of cancers acquired resistance through a missense substitution of D473. The recently reported structure of vismodegib-bound Smo reveals that this mutation disrupts a network of hydrogen bonds that coordinates the drug in the binding pocket [25]. Other mutations that directly impair drug binding include H231R, W281C, V321M, I408V, C469Y, Q477E, and E518K [93]. T241M, A459V, F460L and G497W resistance mutations are distal to the binding site and exert negative effects on drug binding in an allosteric manner [93,103]. For example, molecular dynamics simulations have shown that the region surrounding the mutated residue in Smo-G497W undergoes a conformational change that narrows the drug-binding cavity of the receptor, thereby reducing the inhibitory effect of vismodegib [103]. Activating mutations L412F, W535L, and S533N confer vismodegib resistance by constitutively activating the receptor and hence desensitising it to the drug (Figure 7b).

Although the structure of sonidegib-bound Smo is yet to be determined, the binding sites of sonidegib and vismodegib are likely to be overlapping since they were derived from the same natural Smo inhibitor cyclopamine. It is thus not surprising that mutations that confer resistance to vismodegib also affect responsiveness to sonidegib. For example, a small-cohort study has shown that BCC patients that acquired resistance to vismodegib through the D473H/G and Q477E mutations have a poor response to sonidegib. In the same study, W535L and S533N oncogenic mutations also impaired sonidgeib efficacy [104]. Yet, certain mutations have been identified in mice models that are unique to sonidegib resistance. These residues are conserved in humans and correspond to N219D, L221R, D384N, S387N, and G453S (Figure 8) [46]. Interestingly, there is a record of the G453S mutation in *SMO* in the COSMIC database, suggesting that human tumours can also acquire resistance specifically to this drug.

To investigate whether natural genetic variation can cause unresponsiveness to vismodegib and sonidegib, Smo mutations conferring resistance to these drugs were searched in the GPCRdb. It appears that three vismodegib resistance mutations (D473N, W281C, and I408V) and one of the sonidegib-specific mutations (S387N) are indeed present in the healthy population. This observation highlights the need for genetic profiling of tumours before a Smo antagonist is prescribed. This is because early detection of these pre-existing resistance mutations will likely facilitate better treatment choice and, as a result, improved therapy outcome.

## 6. Discussion

The functional and structural characterisation of Smo has been advancing at a remarkable rate since it was first established to be an oncoprotein in BCC. The mechanisms of Smo activation remained a mystery until the high-resolution X-ray and cryo-EM structures revealed the receptor’s capability to become active upon binding of lipid-based agonists. Alongside these findings, it was recognised that Smo is a highly druggable protein with multiple binding pockets and a range of conformational states, which can aid the discovery of novel synthetic inhibitors. Moreover, Smo-mediated Hh signal transduction can modulate cancer hallmarks through both the canonical and non-canonical pathways as well as crosstalk with other signalling networks.

As the clinical practice is moving away from the one-fits-all approach, cancer treatments tailored towards the genetic characteristics of a patient’s tumour will have a particularly major impact in the future. Although the importance of polymorphisms of other GPCRs in precision medicine has been examined, this review argues that Smo variants should be considered in individualised oncology approaches. For patients whose tumours harbour oncogenic mutations in *SMO*, early detection of these genetic alterations can match these individuals with a suitable Smo inhibitor. This would particularly benefit patients with BCC, BCN, medulloblastoma, meningioma, and HCC since these tumours are Smo-dependent. Indeed, genomic analyses have already been successful in identifying patients with Hh-driven medulloblastoma who would benefit from therapy involving Smo antagonists [105]. Additionally, according to the COSMIC database, oncogenic Smo mutations may also be present in tumours that develop in the large intestine, cervix, oesophagus, upper aerodigestive tract, and bones (Figure 9). To determine whether Smo mutants drive cancer in these tissues, these tumours should be subjected to genomic testing and extensive molecular analysis. Such gaps in knowledge could soon be addressed, as current genotype-directed clinical trials taking place in Denmark and Finland are investigating the use of vismodegib in treating various types of advanced solid tumours [106,107].

The analysis of data from the GPCRdb has demonstrated that Smo mutations that confer resistance to vismodegib and sonidegib are present in the healthy population. This finding highlights the importance of genomic tumour profiling at the earliest stages of cancer to allow the elimination of ineffective treatment with these drugs. The high incidence of acquired resistance emphasises the significance of regular tumour genetic testing throughout the course of treatment. Implementation of this practice would allow effective salvage therapies to be offered in a timely manner. For example, itraconazole is believed to be a good candidate for second generation therapy against Hh-driven cancers [108]. Additionally, combinatorial approaches targeting both Smo and downstream signalling molecules have been recognised as promising in cancer treatment [109]. Targeting regulatory mechanisms of Smo presents another therapeutic approach for acquired resistance. Blocking GRK2 could prevent the essential phosphorylation of the Smo C-tail and dampen the receptor’s activity as shown to be the case in mammalian cells in vitro [110]. Inhibition of cholesterylation is also an interesting direction for drug development since this type of modification is uncommon.

While this review gives an outline of the important oncogenic and resistance mutations, many other mutations have an unknown effect on Smo activity. For example, V270I occurs relatively frequently in both healthy individuals and cancer patients (Figure 6a). Given that this residue is in the important 7TM domain, it is a question of future research to investigate the effect of this mutation on the Smo receptor. In addition to the classical biochemical experiments, the impact of mutations on Smo function should be examined using recently developed structural bioinformatics methods [111]. Furthermore, an understanding of how Smo variants impact drug dosage requirements will be an exciting area of future study as it will spare patients from exposure to ineffective amounts of medication.

As genomic profiling of tumours is becoming part of routine care, oncogenic mutants of Smo must be added to the cancer biomarker toolkit. Further characterisation of Smo variants and their role in various types of tumours will help develop novel therapies with antiresistance profiles and personalised approaches. This will be essential in ensuring improved long-term prognosis for cancer patients.

## Figures and Tables

**Figure 1 jpm-12-01648-f001:**
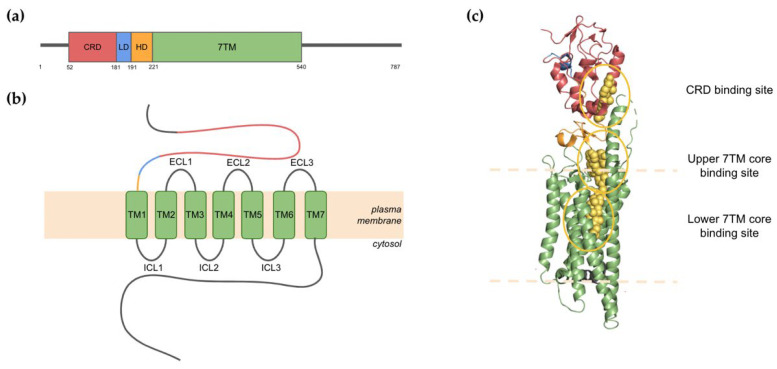
An overview of the structure of the human Smo receptor. (**a**) Domain architecture of full-length Smo; (**b**) A schematic diagram showing the distinct transmembrane regions of Smo; (**c**) Smo is activated by direct binding of small lipid-based molecules. The surface-cartoon representation of human Smo (PDB: 6O3C) highlights three ligand-binding pockets. Cholesterol molecules occupy the CRD and lower TMD binding sites. A synthetic agonist is in the upper binding pocket of the TDM core.

**Figure 2 jpm-12-01648-f002:**
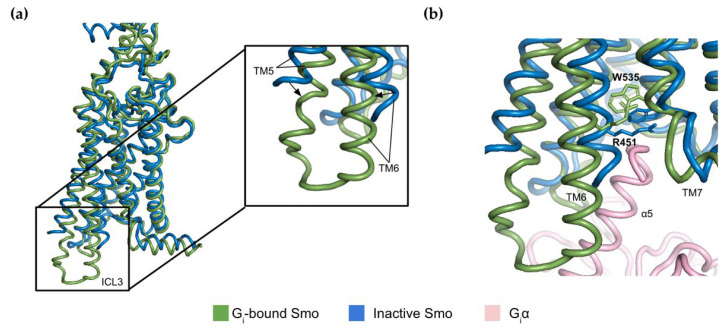
Structural comparison of active and inactive conformations of Smo. (**a**) Superposition of active (PDB:6OT0) and inactive (PDB: 5L7D) structures of Smo 7TM with a zoomed-in view of the structural movements of TM5 and TM6. The fusion protein inserted between TM5 and TM6 of the inactive structure was removed for clarity; (**b**) Differences in relative positions of R451 and W535 between the two conformational states.

**Figure 3 jpm-12-01648-f003:**
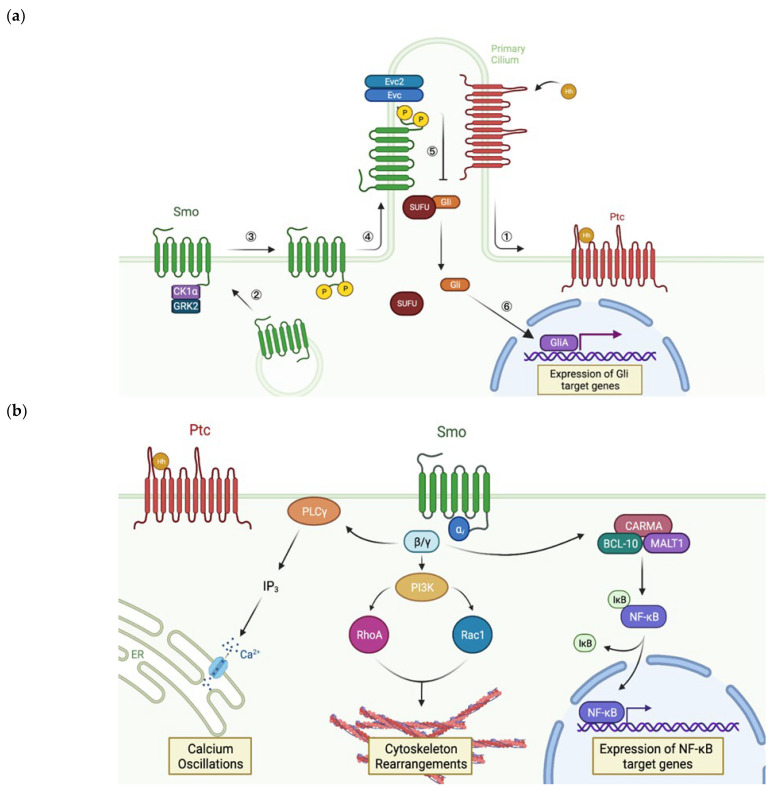
Smo-dependent Hh signalling (created with BioRender.com (accessed on 28 September 2020))**.** (**a**) The canonical pathway. Binding of Hh ligand to Ptc removes Ptc from the primary cilium ① and releases Smo from inhibition. This stimulates accumulation of the receptor on the surface membrane due to increased trafficking of Smo-containing intracellular vesicles ②. The C-tail of Smo is then phosphorylated by CK1α and GRK2 ③, which facilitates ciliary accumulation of the receptor ④. In the primary cilium, Smo interacts with Evc/Evc2. This releases Gli from inhibition by SUFU ⑤. Gli moves into the nucleus, where it gets converted into GliA ⑥. GliA activates transcription of the Hh target genes. (**b**) Type II non-canonical signalling. Hh binding to Ptc releases inhibition of Smo, which allows it to interact with G_i_. This triggers dissociation of G_i_ into ⍺ and β𝛾 subunits. The latter activates PLC𝛾. This leads to production of IP3, which triggers calcium spikes through the opening of Ca^2+^ channels in the ER. Gβ𝛾 also activates PI3K. This stimulates RhoA and Rac to trigger cytoskeleton rearrangements. Finally, Gβ𝛾 stimulates assembly of the CARMA-BCL-10-MALT1 complex, which stimulates the release of NF-κB from its inhibitor (IκB). NF-κB translocates into the nucleus, where it induces expression of its own target genes.

**Figure 4 jpm-12-01648-f004:**
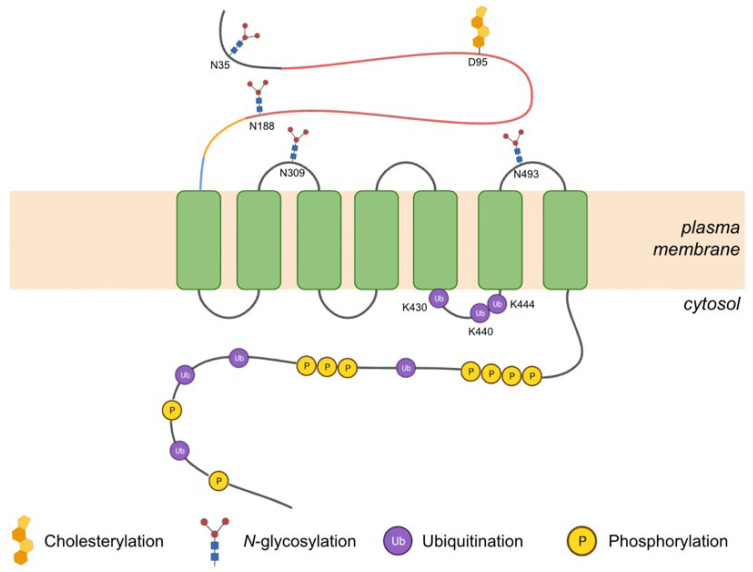
Regulation of Smo by post-translational modifications. The cartoon illustrates functional domains of Smo and locations of post-translational modifications, including the key residues that, if mutated, may alter receptor activity.

**Figure 6 jpm-12-01648-f006:**
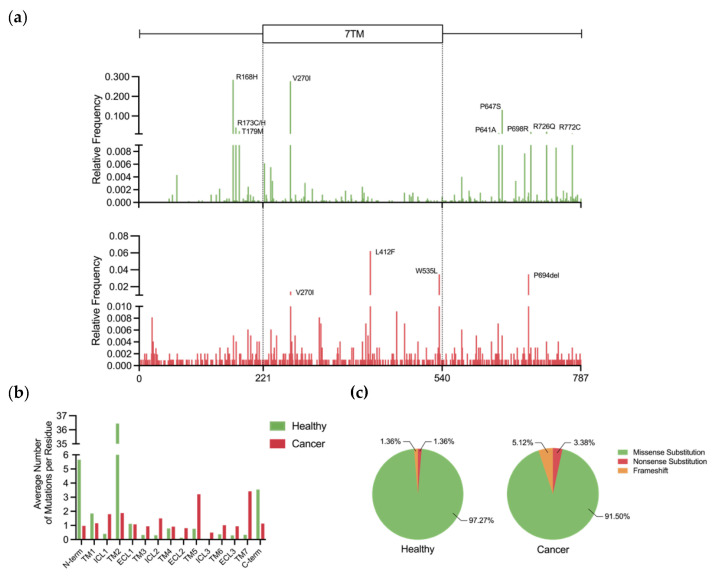
Comparison of *SMO* mutations in healthy individuals and cancer patients. (**a**) The mutational landscape of *SMO* in healthy individuals (green) and cancer patients (red). Relative frequency was calculated by dividing the frequency of each mutation by the total frequency. Labels indicate the most frequent mutations (relative frequency ≥ 0.01). A simplified schematic organisation of Smo domains is shown above. (**b**) Comparison of mutation distribution across the distinct structural domains of 7TM and the flanking segments (denoted as N-term and C-term), corrected for domain length. In cancer, the mutational frequency is higher in the 7TM, except for TM2. In healthy individuals, regions N- and C-terminal to the 7TM are more enriched in mutations than in cancer tissues. (**c**) Pie charts show the abundance of different types of non-silent mutations in healthy (left) and cancer (right) tissues.

**Figure 7 jpm-12-01648-f007:**
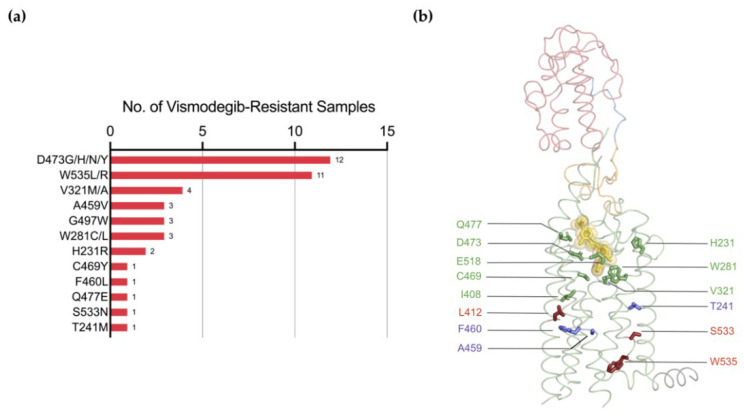
Vismodegib resistance mutations. (**a**) A histogram showing the number of vismodegib-resistant cancer tissue samples with a particular mutation; (**b**) Crystal structure of vismodegib-bound Smo (PDB: 5L7I) with residues that cause drug resistance shown as sticks. Mutations in green residues directly disrupt drug binding. Residues in purple are distal to the binding site and, if mutated, impair drug binding in an allosteric manner (G497 is not shown because EL3 is not observed in the crystal structure). Residues highlighted in red are also outside the binding pocket. Mutations in these residues are oncogenic and alter drug binding by stabilising the active state of the receptor.

**Figure 8 jpm-12-01648-f008:**
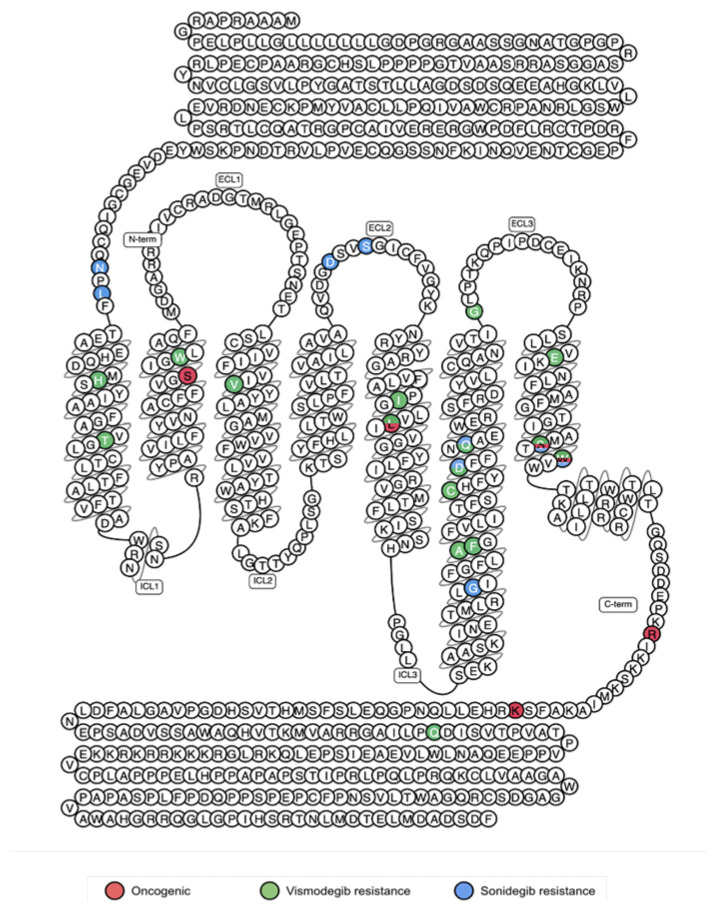
Smo snake plot highlighting locations of functionally important mutations in cancer. The plot shows locations of mutations that drive cancer (red) and cause resistance to vismodegib (green) and sonidegib (blue). Certain residues have dual functions in tumorigenesis and resistance, as shown by colour overlay (modified from www.gpcrdb.org (accessed on 13 February 2022)).

**Figure 9 jpm-12-01648-f009:**
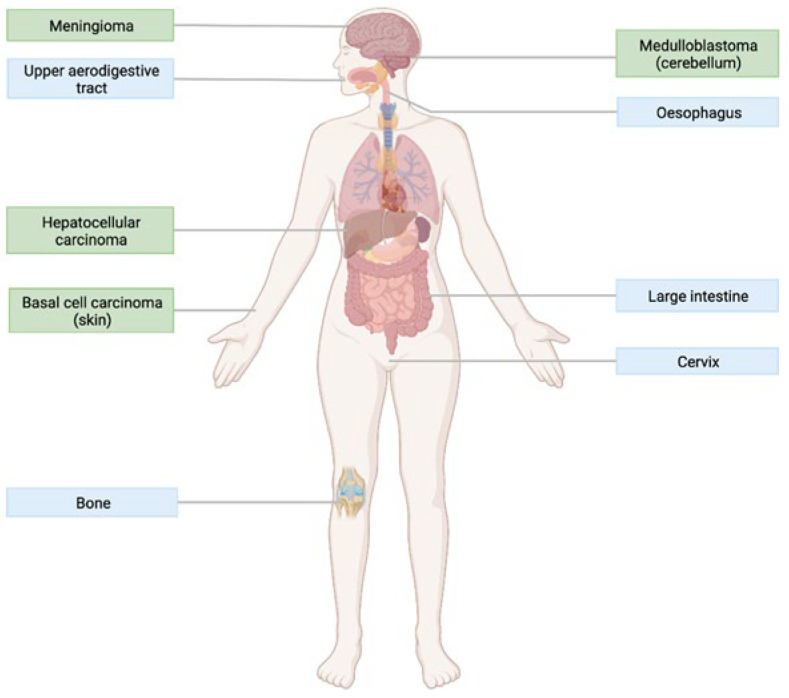
Tissues where activating mutations in *SMO* have been identified according to the COSMIC database (created with BioRender.com (accessed on 20 March 2022)). Green boxes indicate primary tissues where tumours have been confirmed to be driven by Smo. Blue boxes indicate tissues where oncogenic mutations have been detected but it is unknown whether Smo is essential to tumour growth in these tissues.

**Table 1 jpm-12-01648-t001:** Smoothened antagonist clinical trials (completed and ongoing), as listed on clinicaltrials.gov in August 2022.

Drug	Indications	Stage
IP-926 (Saridegib)	Metastatic pancreatic cancer (NCT01130142) Myelofibrosis (NCT01371617) Chondrosarcoma (NCT01310816) BCC (NCT01609179, NCT02828111) BCN (NCT02762084, NCT03703310)	Phase II, III
BMS-833923/XL139	BCC and BCN (NCT00670189) Chronic Myeloid Leukaemia (NCT01218477) Small Cell Lung Cancer (NCT00927875) Gastric and Oesophageal Adenocarcinomas (NCT00909402) Multiple myeloma (NCT00884546)	Phase I, II
LY2940680 (Taladegib)	Advanced solid tumours (NCT02784795) Small Cell Lung Cancer (NCT01722292) Gastroesophageal Junction Adenocarcinoma (NCT02530437)	Phase I, II
Vitamin D3	BCC (NCT01358045)	Phase II
Itraconazole	BCC (NCT02120677) Non-Small Cell Lung Cancer (NCT02357836) Prostate Cancer (NCT01787331) Oesophageal Squamous Cell Carcinoma (NCT04018872)	Phase II
LEQ-506	Advanced solid tumours (NCT01106508)	Phase I

**Table 2 jpm-12-01648-t002:** A summary of Smo variants that have been identified to drive cancer progression in different Type I tumours.

Tumour	Driving Mutations
Basal cell carcinoma	R562Q, W535L
Basal cell nevus syndrome	L412F
Hepatocellular carcinoma	K575M
Medulloblastoma	W535L, S533N, S278I
Meningioma	L412F, W535L

## Data Availability

Protein crystal structures are available on the Protein Data Bank (https://www.rcsb.org/ (accessed on 13 February 2022)) under the accession codes specified throughout the manuscript. Analysis of mutation prevalence and incidence was performed using data from the GPCR database (release of 28 July 2022; https://gpcrdb.org/ (accessed 30 July 2022)) and the COSMIC database v96 (https://cancer.sanger.ac.uk/cosmic (accessed on 30 July 2022)).

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
