# Peer review of "Towards Precision Oncology: The Role of Smoothened and Its Variants in Cancer"

_jpm, 2022, doi:10.3390/jpm12101648_

Round 1
Reviewer 1 Report
In this review, the authors discussed the Smoothened and its variants in cancer. The topic in interesting and well discussed but there are some points in the manuscript that must be revised or completed to be accepted.
1. The words that are used for the first time in the text should not be abbreviated. Unless it is mentioned in parentheses after the main word. For example, Hh on page 1.
2. Figure 3 is confusing. The canonical and non-canonical HH signaling pathways are not clear. This figure needs to be revised.
3. Mention different variants of Smo in different cancers separately and discuss why some Smo variants are more common in cancers.
Author Response
We would like to thank the reviewer for their evaluation of our work. Below we describe the actions we have taken to address each of the specific comments provided by the reviewer.
- The words that are used for the first time in the text should not be abbreviated. Unless it is mentioned in parentheses after the main word. For example, Hh on page 1.
We have ensured that all words used for the first time in the text have the abbreviation that will be used henceforth mentioned in parentheses after the main word.
- Figure 3 is confusing. The canonical and non-canonical HH signaling pathways are not clear. This figure needs to be revised.
Figure 3 has been amended to more clearly demonstrate that the canonical pathway results from the binding of Hh ligand to Ptc, which removes the inhibition on Smo and allows the phosphorylation of Smo that results in the removal of inhibition of Gli, which is then able to express its target genes. We have also numbered the steps of the canonical pathway to make them clearer for the reader.
- Mention different variants of Smo in different cancers separately and discuss why some Smo variants are more common in cancers.
We have included a new table, Table 2, to show the different cancers and the Smo variants that have been identified to drive tumour progression in each. We have also added the following sentence:
By disrupting the structural integrity of the inactive form of Smo, these mutations activate the Hh signaling and trigger tumorigenesis. For this reason, it would be expected that these mutations would primarily be found in cancer patients.
Reviewer 2 Report
In this manuscript, the authors provide a very comprehensive review of Smoothened. The review is very well written and very complete, discussing the structure and function of smoothened, its role in the hedgehog signaling pathway, its role in cancer, and its potential as a target of directed therapy. I think this comprehensive review will be very well received and I have no comments.
Author Response
We would like to thank the reviewer for their evaluation of our work. We note that the reviewer has recommended publication as is with no revisions needed.
Reviewer 3 Report
Comments for the authors:
This review provides information about the mutational landscape of Smo and its role in cancer. Overall, the manuscript is well-written and organized. However, the following issues should be addressed:
1. The authors described that binding of Hh ligand to Ptc removes from PC and releases Smo inhibition. However, the process was not depicted in Figure 3.
2. The authors could provide a table summarizing the genetic alteration or expression of Smo in human tumors.
3. In section 4.2, the authors mention that as Type I cancers arise independently of the ligand, inhibition of ligand binding to Ptc will have no effect on these types of tumors. Which tumor types are classified as Type I?
4. The authors mention that three Smo antagonists have been approved by the FDA. The mechanism of action of these Smo antagonists could be described briefly.
5. In Table 1, the NCT number should be presented in the table.
Author Response
We would like to thank the reviewer for their evaluation of our work. Below we describe the actions we have taken to address each of the specific comments provided by the reviewer.
- The authors described that binding of Hh ligand to Ptc removes from PC and releases Smo inhibition. However, the process was not depicted in Figure 3.
The figure has been amended to depict this process.
- The authors could provide a table summarizing the genetic alteration or expression of Smo in human tumors.
We have added a table of Smo variants and the types of cancer in which they have been identified in the COSMIC database of human tumours.
- In section 4.2, the authors mention that as Type I cancers arise independently of the ligand, inhibition of ligand binding to Ptc will have no effect on these types of tumors. Which tumor types are classified as Type I?
As we mention in our response to comment 2, we have added a table of Smo variants and the cancers in which they are found. All of these tumors are classified as Type I. We have indicated this in the text and in the Table legend.
- The authors mention that three Smo antagonists have been approved by the FDA. The mechanism of action of these Smo antagonists could be described briefly.
We have added the following sentence: Vismodegib, sonidegib, and glasdegib prevent Smo activation by blocking the binding sites of lipid-based agonists, which are found at increased levels in the plasma membrane in the presence of Hh.
- In Table 1, the NCT number should be presented in the table.
The NCT numbers have been added to Table 1 as requested.